# Equality, Diversity, and Inclusion Strategies Adopted in a European University Alliance to Facilitate the Higher Education-to-Work Transition

Anna Siri [1],*, Cinzia Leone [2] and Rita Bencivenga [2]

1 Department of Education Sciences, University of Genoa, 16128 Genova, Italy
2 Department of Chemistry and Industrial Chemistry, University of Genoa, 16146 Genova, Italy
* Correspondence: anna.siri@unige.it

**Abstract:** The COVID-19 pandemic has disrupted higher education, imposing the need to add new strategies to academic educational models to facilitate young people's transitions from education to work. Among the new challenges, the research study focuses on the importance of valuing and incrementing inclusion, raising awareness of equality, diversity, and inclusion (EDI) strategies and policies. Many universities have yet to develop inclusive processes and cultures that provide equality of opportunity for all, regardless of gender, ethnicity, social class, sexual orientation, physical ability, identity, and cultural background. Since 2019, the European Commission has financed "European Universities", networks of universities creating international competitive degrees that combine excellent study programmes in different European countries. Today, 340 institutions in 44 European University Alliances (EUAs) promote European values and identity and revolutionise their quality and competitiveness to become the "universities of the future". This article proposes a comprehensive approach to promote EDI within the EUA "ULYSSEUS" involving Spanish, Italian, Austrian, French, Finnish, and Slovakian universities through micro-actions to apply EDI principles at the project level. The authors will frame the theoretical basis of the experience through documentary analysis and their academic expertise in promoting strategies connected with the European values enshrined in Article 2 of the Treaty on European Union: pluralism, tolerance, justice, solidarity, non-discrimination and equality. Implementing these values through visible micro-actions could document and counteract the disadvantages underrepresented groups face in academia. In the mid-term, the experience had by the students in the EUA could facilitate the higher education-to-work transition, allowing them to replicate their EDI-related experience as students to their future roles as citizens and workers. The outcome could thus contribute to a life-wide learning perspective for a more inclusive Europe in the long term.

**Keywords:** equality; diversity; inclusion; European University Alliances; gender action plan-GAP; gender equality plan-GEP; P-GAP

## 1. Introduction

The COVID-19 pandemic has disrupted higher education, imposing the need to add new strategies to academic educational models to facilitate young people's transitions from education to employment.

Numerous research studies analyse how universities and organisations have responded to the impact of the COVID-19 pandemic in terms of preparing university students and recent graduates to enter the global labour market, referring to a variety of sectors, such as the accounting, banking, and finance sectors [1], and the digital one, in particular observing the consequences—partially positive—of the forced digital revolution [2]. Worries linked to direct or indirect damages to students' learning paths and future chances in the working environment at large identify groups at risk as working students, parent

students, and students living in remote areas, with the potential increase of inequality-related economic and social aspects [3,4]. The difficulties experienced by students with special needs or disabilities have been observed [5] but also the positive lessons to be learnt regarding accessibility and the possibility for adaptation going forward, for staff and students alike [6].

Other research related to the impact of COVID-19 focused on the work-life balance and, consequently, on many female academics [7]. Additionally, the importance of rethinking the concept of ethics of care in the academic field has been analysed [8]. However, this research strand focused on academic and even administrative staff, only marginally including the impact on students.

The link between COVID-19 and EDI was addressed by describing academic initiatives at department level, often in the health sector [9,10]. These initiatives focus on creating committees or pledges, and the articles analyse their impact on faculty, staff, and trainees. Also in these cases, the transition between education and work is not addressed directly.

At the academic level, clear indications have been issued by reports and position papers [11–13] that define the best practices and the approaches to follow to progress rapidly and steadily towards more inclusive academia, supporting institutional growth and capacity building to promote the progress and innovation of European society. Moreover, to be persuasive, institutional leaders are invited to adapt their arguments for diversity and inclusion to different audiences and contexts [14].

In the following pages, the research study will commence with a mapping of universities' strategies on EDI, before moving on to a short description of the European Universities Initiatives, in which the case study is embedded. The paper will then focus on how micro-actions on EDI are being promoted in an EUA of six European Union (EU) partners, before making some concluding comments about the exploitation potential of this approach in the post-COVID-19 pandemic academic environment. After discussing limitations of the work conducted, areas for further research will be identified.

## 2. Equality, Diversity and Inclusion and Higher Education

EDI is a strategic topic for the higher education sector. It impacts institutional culture, research, and teaching, implementing policies that affect professional and personal activities. Universities strive to develop inclusive processes and cultures that provide equality of opportunity for all, regardless of gender, ethnicity, social class, sexual orientation, physical ability, identity, and cultural background.

Higher education ministers at the Ministerial Meeting of the Bologna Process in 2015 agreed to make higher education systems more inclusive. This was reiterated by the European Commission in its 2017 renewed agenda for higher education. Other more indirect strategies are having an impact on EDI in the academic sector, for example the Horizon Europe request for all public entities that will apply for funding to have a Gender Equality Plan and adopt a Gender+ strategy in which gender remains the main contemplated type of inequality. However, its interaction with other sources of inequality and grounds of discrimination is taken into account in the design and implementation of the GEP measures, as well as intersectional indicators. This request promotes initiatives not targeted only to gender but with a wider EDI perspective.

Reports and position papers [12,13] in the academic sector define the best practices and the approaches to follow to progress rapidly and steadily towards more inclusive academia, supporting institutional growth and capacity building to promote the progress and innovation of European society. A book promoted by the Council of Europe reaffirms that to be persuasive, academic institutional leaders need to understand how to adapt their arguments for diversity and inclusion to different audiences and contexts [14].

Inclusiveness is therefore a strategic question for the higher education sector, as it impacts institutional culture, research and learning and teaching. Higher Education Institutions aim to be more open and inclusive and find new ways to enable people from

traditionally less-represented backgrounds to participate and progress in their working or learning careers, thus increasing diversity.

Since the early 1960s, diversity management has been commonly focused on historically disadvantaged groups such as women and minorities, but the concept of diversity has expanded over time, due to growing awareness about differences. Currently, sexual and gender diversity, age and other grounds for potential discrimination have become more visible. The grounds for potential discrimination recognised by EU legislation, in the EU Charter for fundamental rights, are sex, race, colour, ethnic or social origin, genetic features, language, religion or beliefs, political or any other opinion, membership of a national minority, property, birth, disability, age or sexual orientation. To promote EDI in academia it is important to monitor these grounds. The challenge is facilitated by the focus placed by the European Commission on addressing equality, diversity and inclusion in an intersectional [15] perspective, stimulating the progress towards inclusive organisational practices that foster equity across multiple intersecting identities. An approach focused on the political perspectives of higher education is offering new outlooks worthy of attention when implementing EDI practices [16].

The League of European Research Universities (LERU) remarks that "One of the impediments to sustainable change is that many of the initiatives to promote equality, diversity and inclusion at universities have not been joined-up. The researchers identify three gaps in these approaches that need to be closed: (1) Efforts have not been sufficiently synergistic in tackling the common barriers faced by all under-represented groups, including women, ethnic or cultural minorities, LGBT+2, people with disabilities or first-generation members of the university community; (2) The focus of many efforts has been either on staff issues, or on student issues, rather than on addressing the needs of the university community as a whole; (3) The value of building inclusivity into the teaching curriculum or the design of research and innovation programmes (e.g., taking account of how considerations of sex/gender and/or minority perspectives could impact the research questions, methods and processes have often not been sufficiently central to efforts to address institutional issues of diversity and inclusion" [13]. However, increasing the diversity of students in HE cannot be taken as an indicator of greater 'equality' within the system, or of an equitable HE system: "this focus upon essentialised, individualised bodies detract attention from the role of the culture and organisation of higher education, and the impact that these have on 'equality' and 'diversity' issues" [17] (p. 646).

Among the main actors, students should be included. "Any comprehensive effort to promote diversity, access and inclusion on campus must also acknowledge and harness the energy and participation of the students themselves. Even if diversity-related efforts are well intentioned, they are likely to be doomed to fail in the long run if they are perceived by students to be completely top-down. For this reason, higher education institutions must empower students and student organisations to play leadership roles in facilitating dialogue across differences" [18] (pp. 71, 72).

Research shows the little engagement of students with disabilities, non-academic staff, and academic staff in the academic literature focused on EDI and on students with disabilities [19]. Since the beginning of the COVID-19 pandemic, a widening of inequalities, at least in STEMM (science, technology, engineering, mathematics and medicine) academic disciplines, has been noticed, affecting people who are already marginalised within these fields [20]. These warnings deserve further attention and monitoring. However, while in the last two years remote working and studying has lowered the number and quality of interactions, and difficulties for people at risk of marginalisation have been documented, online networks by students' activists belonging to marginalised groups or groups at risk of discrimination have flourished, [21] and social media could contribute to nurturing the awareness of EDI-related principles.

### 3. Higher Education and Work Transition

The school-to-work transition is a process that starts in the years of higher education and ends when graduates have found adequate employment [22]. Students need to acquire the necessary competencies enabling them to adapt to the dynamic workplace and continuously develop themselves [23–25].

Higher education and the world of work are rather disconnected, and since 1999, with the Bologna declaration, the EU initiated a change from the delivery of knowledge towards putting knowledge in the context of students acquiring competences. This has influenced the emergence of competence-based education [25–27], expected to better prepare students for their transition to the labour market and their professional future [25–28]. Taking a competence-based approach, coaching practices have been developed to enhance students' employability competences to facilitate the transition [25].

Eurostat data highlight the impact of the COVID-19 crisis for young people (15–29 years). Figure 1 compares Q3 2021 with Q3 2019 (pre-pandemic level). The same quarter (Q3) is taken for the three years as no seasonally adjusted data are available for the specific age group 15–29. The data show that all countries experienced a decline in the youth employment rate. Most EU member states (16 out of 27) still did not show a full recovery at the end of Q3 2021, as the youth employment rate was still below the rate recorded in Q3 2019. The largest decreases were recorded in Portugal, Bulgaria, Latvia, the Czech Republic, and Poland.

In contrast, the largest increases in the share of employed 15–29-year-olds in Q3 2021 compared to the pre-COVID situation were in Ireland, France, and Slovenia.

The issue of young individuals being one of the categories hit the most by the devastating economic impact of the COVID-19 pandemic has been acknowledged in the literature [28–30].

The transition between HE and work is characterised by individual differences and micro and macro contexts [31,32].

Consensus is lacking regarding which competencies are more relevant; however, the importance of a set of skills that goes beyond specific knowledge to deal with the complex and rapidly changing demands of a globalised world emerges clearly [33,34]. This set of resources is commonly designated as transversal competencies. These factors that facilitate or constrain the transition to the labour market are particularly appreciated in a "period of turbulence and continuous changes in the labour market" [35]. In this context, it is of paramount importance to add to those transversal skills and competencies the personal and direct experience had towards EDI during the students' years. Academic and student collaborative networks dedicated to EDI [36] would represent a decisive and innovative change agent in our academies. Sobrany et al. [36], in fact, refer to a successful case study related to the creation of an advisory group within the academy, based on EDI values.

Although the focus of the article is not on youth policies, it is worth mentioning the temporal and social dimensions analysed by Andy Furlong, through the observation of the relationship between continuity and change in young people's lives and the place of youth in the reproduction of inequality across generations [37]. The difficulty in capturing the cultural diversity of modern youth risks leading to framing "youth policy in generalised terms, underpinned by assumptions of homogeneity and linearity and viewed through the lens of an older generation who grew up in very different times" [38] (p. 367). European Universities can and will influence future youth policies, and the potential generational clashes, particularly in countries like Italy, where the average age of academic personnel is relatively high, could soon become a reality.

Another aspect deserving attention is that with the massification of HE in Europe, HE students are often conceptualised "as people undergoing a series of potentially transformative changes" [39] (p. 24). Brooks et al. show to what extent attending university may be seen by the students as a rite of passage and examine two common ways in which the idea of being 'in transition' is seen: as preparation for the labour market and as a personally transformative experience. Moreover, the research confirms that national traditions of HE, cultural norms, and other socio-historical factors may all affect how the construction of

students as 'in transition' is understood and potentially lead, the authors add, to consequences on the transition to work. This article describes activities that could contribute to facilitating the transition from HE to work through acquiring awareness and experiences of EDI practices. The experience is currently implemented in a European University Alliance as a possible driver for change at the European level, within similar alliances in Europe, and at the academic level in general.

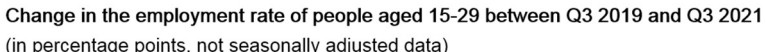

**Change in the employment rate of people aged 15-29 between Q3 2019 and Q3 2021**
(in percentage points, not seasonally adjusted data)

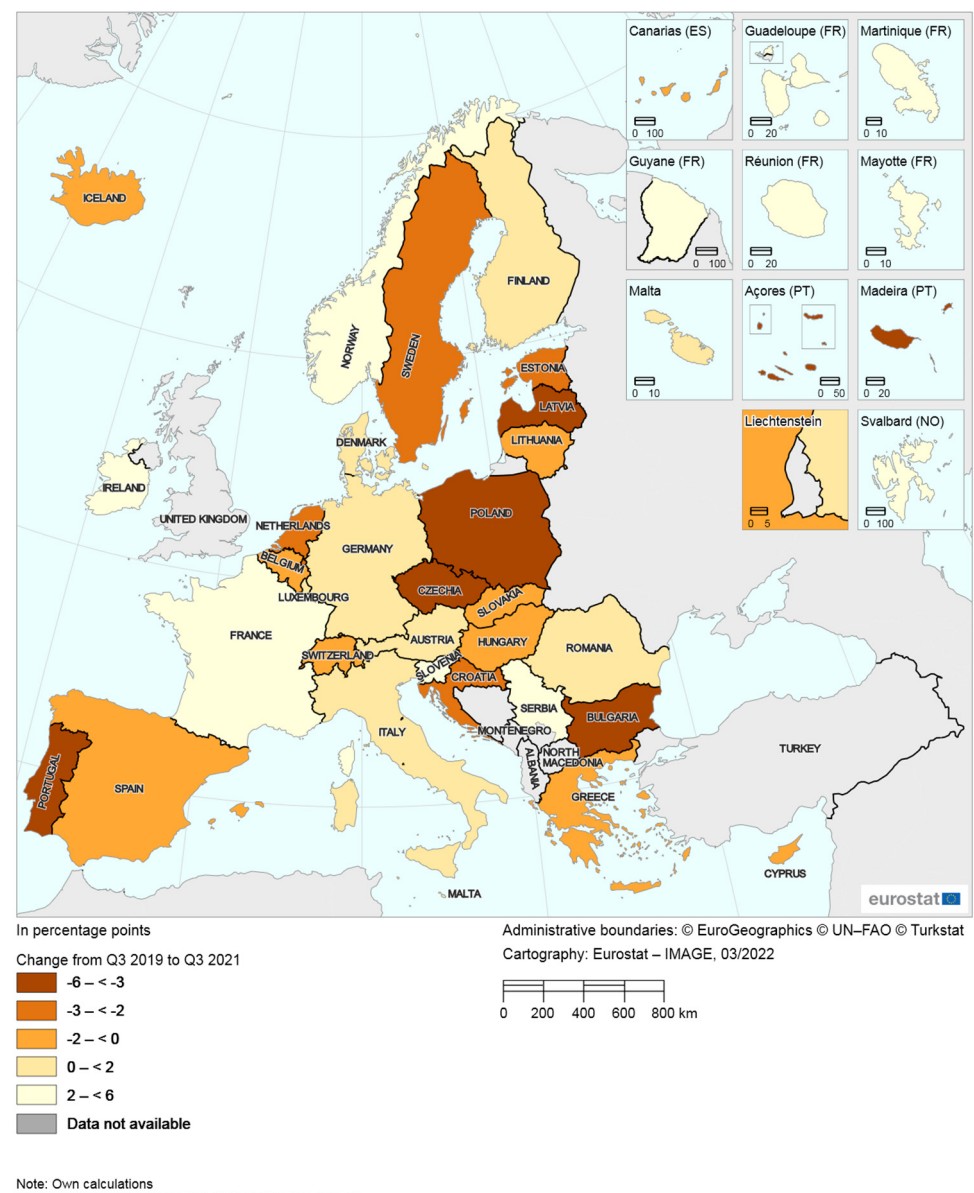

**Figure 1.** Impact of the COVID-19 crisis for young people (15–29 years). Source: Eurostat (lfsq_pganws) and (lfsi_emp_q).

## 4. The European University Initiative and ULYSSEUS

In 2017, the European Council invited EU member states to continue actions to strengthen partnerships between higher education institutions across the EU. In particular, it was deemed important to encourage the emergence by 2024 of "European Universities", consisting of bottom-up networks of higher education institutions across the EU to offer

students the opportunity to earn a degree by combining studies in different EU countries, thus contributing to the international competitiveness of the European academic sector [40].

Later in 2018, moving toward a vision of a European Education Area (EEA), the Council further recognised the leading role that "European Universities" could play in supporting future-oriented education and training systems [41].

In their first joint meeting in 2019, ministers of education and finance stressed the need to step up effective and efficient investment in education and training, skills, and competencies in terms of quality and quantity, inclusiveness, and equity. In this, the meeting anticipated what the Council recognised in the following year on education and training that the investment in education and training is the most potent investment in people and the future, and offers social and economic returns for individuals, employers, and society as a whole [42].

As highlighted in the European Skills Agenda, adopted by the Commission in 2020, and in other resolutions [43–45], "European Universities" are considered a flagship example for modern and inclusive higher education institutions of the future in Europe in the way they promote gender equality, inclusiveness, and equity.

As a result of two Erasmus+ calls for proposals, complemented by support from Horizon 2020, 41 pilot "European Universities" alliances are now working towards their shared vision and institutional change, aiming for a structural, systemic, and sustainable impact on education and training, research and innovation, and service to society. They involve more than 280 higher education institutions, covering 5% of all higher education institutions in Europe, with the potential to affect 20% of European students.

European labour markets are rapidly changing due to technological development, digital and green transitions and restructuring of the economy, but the COVID-19 pandemic also influences them. The importance of flexible learning formats at all stages of life, innovative curricula, flexible arrangements, and alternative learning pathways emerges to improve quality, inclusion, and gender equality in higher education [36].

The COVID-19 pandemic has demonstrated that deeper cooperation across borders, disciplines and cultures is the only way to recover from the crisis and become more resilient. These "European Universities", therefore, represent a crucial element of the European educational landscape.

The "European Universities" initiative is an example of how Europe relies on highly integrated complementary promotion programs that focus on synergies between education, research, and innovation. The initiative provides new systemic, transformative impulses to academic education by raising quality standards and increasing the competitiveness of the entire European higher education area.

European universities are well placed to drive a sustainable impact on the broader higher education sector in Europe and worldwide. They can accomplish this result through their mission to lead education, research and innovation in a common direction that supports Europe's knowledge as the basis for democratic, resilient, and inclusive societies, and the core European values of academic freedom, openness, and scientific integrity.

This article is based on a research work in progress within the European University Alliance called "ULYSSEUS—An open to the world, people-centred and entrepreneurial European University for the citizenship of the future" (https://ulysseus.eu, accessed on 1 October 2022), realised with the financial support of the Erasmus+ and Horizon 2020 programs. ULYSSEUS, over the next three years, involves the University of Genoa (UniGE) together with five other European universities: University of Seville (Spain), Université Côte d'Azur (France), Technicka Univerzita v Kosiciach (Slovakia), Management Centre Innsbruck (Austria) and Haaga-Helia University (Finland).

The ULYSSEUS alliance integrates four European regions and six different universities, firmly rooted in their territories, sharing common goals and lasting successful cooperation. From North to South, from comprehensive universities to specialised universities, from established and research-based universities to experts in entrepreneurship and academic innovation.

The difference is ULYSSEUS' main strength to create a dynamic and versatile alliance, ready to develop a responsive, sustainable, and proactive community essential to boost the four missions of the higher education institution (education, research, innovation, and service to society) through a spatial and digital Innovation Ecosystem.

In this sense, the six partners of the alliance aim to contribute to its competitiveness, strengthen its innovative character and improve the employment capacity of citizens in their respective areas of influence, as well as to promote among its students and employees an active exercise in citizenship, social inclusion, cohesion, and personal development. To this end, the project has the support of more than 95 partners, including local councils, regional governments, companies, think tanks and NGOs.

ULYSSEUS activities will be developed in a digital and regional campus that will include research centres, business incubators, living labs and open classrooms, as well as a digital platform for whole community interaction. To this end, the alliance has established six "innovation hubs" aligned with six research and development challenges prioritised by the partners' regions and cities: Ageing and wellness, energy, transportation, mobility and smart cities, tourism, arts and heritage, digitisation, artificial intelligence and food, biotechnology, and the circular economy.

The European challenge to internationalisation and modernisation of education systems promoted by "ULYSSEUS" aims to generate a long-term alliance, with a unified structure and strategy for education and research, linked to the priorities of the cities and regions of the universities involved in the project.

In this context, equality, diversity, and inclusion are addressed through an Equality Plan that lays the foundation for the development of the planned initiatives. A work package (WP5) is specifically dedicated to social responsibility and citizen engagement, having as an objective to reach different categories at risk of discrimination due to their gender, age, physical or sensory ability, sexual orientation, and social disadvantage. This research fits into this context in an innovative way, proposing a comprehensive approach to promote equality, diversity, and inclusion within an EUA through micro-actions to apply EDI principles at the project level.

## 5. EDI and Micro-Actions to Favour Higher Education Students' Transition to Work

Before describing the initiative implemented, the authors deem it relevant to explain their roles at UniGe and in ULYSSEUS. This information will allow the readers to understand better why and how the researchers had the idea of linking EDI to the students' transition to work. The authors work in different Departments of the University of Genoa, ULYSSEUS WP5 coordinator, and they share numerous responsibilities for developing and implementing the WP5 Tasks. They are also members of the "UniGe observatory for monitoring the GEP and Gender Budget". In ULYSSEUS, the researchers try to promote EDI inspired by strategies connected with the European values enshrined in Article 2 of the Treaty on European Union: pluralism, tolerance, justice, solidarity, non-discrimination, and equality.

Implementing these values through visible micro-actions could document and counteract the disadvantages underrepresented groups face in academia. The initiative is inspired by the Gender Action Plans project requested by the EU during the 6th Framework Programme for research and Technological Development, and the current request to have a Gender Equality Plan (GEP) to receive funding by Horizon Europe, the EU 9th Framework Programme for Research and Innovation.

The EU 6th Framework Programme for Research and Technological Development (FP6) included three closely gender-related objectives for research: increasing the number of female researchers taking part in projects, ensuring women scientists are involved in the assessment, consultation and implementation processes, and redesigning research to ensure it meets the requirements of both women and men. To achieve these objectives, the European Commission asked the scientific community to include a Gender Action Plan

(GAP) in research proposals for Integrated Projects and Networks of Excellence. GAPs were supposed to contain three key elements:

(1)   Analysis of the situation regarding female participation within the project and gender aspects of the research field;
(2)   Proposed new actions based on the analysis;
(3)   Concrete information as to how the gender dimension would be integrated into the research content during the project.

A survey showed that the FP6 GAPs planned five main categories of measures:

- Quantitative measures for promoting women in projects: recruiting strategies; statistics on participation of women in the project and related monitoring; efforts to increase the numbers of women in management positions; quotas for training courses, workshops, and further training programmes.
- Measures for promoting awareness of gender issues: establishment of gender groups; informational events; monitoring of GAP implementation; collection and publication of statistics; use of gender-sensitive language.
- Measures for networking and exchanges: contacts with women's groups; networking between women scientists; platforms for discussion (Internet, newsletters); events in cooperation with women's groups and other projects; creation of databases on women scientists.
- Measures for promotion of young scientists: contacts to schools and universities; "Girl's Day"; training programmes.
- Measures to promote and facilitate balancing of work and life; customised GAPs at partner institutions; special mobility grants and prizes for women scientists [46] (pp. 4, 5).

In those years, the EU focused on rebalancing the gender participation in science. It is worth noticing that since then the perspective has changed, and the requirements of Horizon Europe, the current Research and Innovation Framework programme, refer to a gender+ strategy, meaning that gender remains the main contemplated type of inequality but its interaction with other sources of inequality and grounds of discrimination must be taken into account in the design and implementation of measures aimed at promoting an advancement in gender equality. Furthermore, it is required to adopt a "gender dimension" in research and innovation, an umbrella that entails considering sex and gender in the whole R&I process. Since January 2022, GEPs have been mandatory requirements for those public bodies, in general, including hospitals, museums, and research organisations or higher education institutions, established in a Member State or Associated Country, submitting proposals to the Horizon Europe programme. In addition, many private organisations have some form of document for promoting gender equality and respect for diversity, for example the activities on Microsoft's Global Diversity and Inclusion, diversity, equity, and inclusion (DE&I) at McKinsey. The researchers are aware of the debate about the ideologies of neoliberalism that have made education policy fit the market's requirements by accommodating sectorial interests and the call for re-establishing the "public" in education [47,48]. However, in the HE-to-work transition, the similarity of initiatives that promote gender equality and diversity in the higher education and corporate sectors constitutes a potential advantage if openly tackled with students.

GEPs are at institutional level and must fulfil mandatory process-related requirements and address a minimum of five content areas: work-life balance and organisational culture; gender balance in leadership and decision-making; gender equality in recruitment and career progression; integrating the gender dimension into research and teaching content and measures against gender-based violence, including sexual harassment.

When the obligation of providing a GEP to qualify for funding under Horizon Europe was announced, the Taskforce from the Work Package 5, composed by at least one representative of each university involved in ULYSSEUS, aimed at creating a Project Gender Equality Agenda, debated whether and how to integrate the request into its working plan.

Not all ULYSSEUS Universities had a GEP at this stage, and Taskforce participants felt that reciprocal support was needed to reach this goal.

The work to be done to promote gender equality, diversity, and inclusion in ULYSSEUS was therefore reshaped, following the new Horizon Europe concept of gender+ and the new request to apply a gender dimension, so that it could be integrated, where possible, into future GEPs. Since the activities were to be implemented at project rather than organisational level, as it was the case in FP6, the FP6 GAP model was considered more appropriate to the scale of the work within ULYSSEUS. The focus is on actions that were realistic at project level.

## 6. Methodology

The micro-actions were described following the general recommendations applied when writing a GEP. The template adopted was inspired by the H2020-funded SAGE project [49], which provided a detailed outline for creating a GEP, which has been simplified to facilitate the work of the partners.

The project-level Gender Action Plan (P-GAP) is transversal to the whole project and can therefore influence all its activities.

Typically, the fertilisation of all project areas begins not with the central project structure, coordination, and management, but with a work package on social inclusion, based on a bottom up rather than top-down approach.

To identify the micro-actions, a qualitative research technique, the focus group, was chosen because of its technical potential, which derives from the exploratory capabilities inherent in the interactive, verbal, and non-verbal communication of small groups. The focus group is, in fact, able to emphasise the objective of drawing out the expertise and opinions on EDI from each participant, through a constructive comparison. The group consisted of the ULYSEEUS WP5 teamwork.

In the study presented, the objective consisted in the detailed description and knowledge of the phenomenon investigated (EDI within the alliance and possible developments) and there was no interest in generalisations. The pursuable goal was not to lead the group toward decision-making, nor to seek consensus on a topic.

In the end, the taskforce participants chose the micro-actions from an initial list of possible actions provided by the Task leader, adding others that were deemed relevant at local level or significant at consortium level.

## 7. Results

While the ULYSSEUS GAP has a wider aim, involving administrative, teaching and research staff, it has been deemed important to include students in two sets of actions:

(1) Promoting a more significant involvement of students in the bodies and initiatives already active in partner universities and harmonised through the collaboration promoted by ULYSSEUS. As an example, and due to page restrictions, only the initiatives and bodies available at the Italian Partner institution, UniGE, are listed (Table 1), which are similar to those of the other five partners. UniGE has two statutory bodies devoted to protecting gender equality and contrasting discrimination: the CUG (Comitato Unico di Garanzia), which was established pursuant to law number 183/2010 and provided for in article 28 of the University By-Laws, and the CPO (Comitato per le Pari Opportunità), established in accordance with article 27 of the University By-Laws. The CUG and the CPO have collaborated to draw up the GEP using existing available data and the actions they plan to take, in particular, the Positive Action Plan 2021-2023 and the Gender Budget (GB) 2019 and 2020. It is worth mentioning that the role and the function of the Guarantee Committee for Equal Opportunity, Employee, Well-being, and Non-discrimination at Work (the CUG's full name) is to promote and strengthen equality between workers of the public sector, especially, but not only, on the grounds of gender. Therefore, its activities, and the PAP it produces every three years, have a scope closer to EDI than the GEP, focused mainly on gender equality and partially on promoting diversity. This broader approach has been

kept in the micro-actions developed at UniGE and implemented in ULYSSEUS, adding a gender+ perspective closer to EDI to the P-GAP plan.

**Table 1.** EDI-related activities.

| Starting Date | EDI-Related Activities That Impact on Students' Lives and Careers |
|---|---|
| 2015 | Agreements have been in place with institutions to support members of staff taking care of individuals who are not self-sufficient |
| 2015 | Activated a double student university ID allowing gender-transitioning students, upon their request, to obtain a temporary bureaucratic profile ("carriera alias") and a new student university ID bearing the student's name of choice. |
| 2017 | Consigliere/a di fiducia (Harassment advisor), with a three-year mandate. A professional practitioner from outside the university trained to offer specific support to anyone in the academic community who is seeking assistance in matters of harassment and/or mobbing. |
| 2018 | The code of conduct for the prevention of any form of discrimination, harassment, mobbing |
| 2018 | A parents' fund for specific categories of non-tenured teaching staff (research scholarship holders but also PhD students and specialising students). |
| 2021 | Delegate for equal opportunities and inclusion) |
| 2021 | The procedure aiming at protecting whistle-blowers |

Source: Own elaboration. Citations are included within the text and are linked to a reference containing the all the information needed to locate the original document.

Tables 1–3 were developed from each partner's Gender Equality Plan. The information was compared, and the most suitable ones were identified. Based on the expertise and experience of the research team, additional actions were then added to the case study.

**Table 2.** The three major strategic approaches to gender equality in science research, policy, and practice.

| EU Recommendations | Micro-Actions in ULYSSEUS | Micro-Actions Involving Students and ESR in the P-GAP |
|---|---|---|
| Fix the Numbers: focuses on increasing women's and other under-represented groups' participation. | To reach a gender balance (40/60) in ULYSSEUS and to increase women's and other under-represented groups' participation to workgroups, project commissions, committees, events, etc. in case of an unjustified imbalance | When involved in events or attending conferences, workshops seminars or any training event, observe the level of diversity among the speakers and participants. Be prepared to ask for explanations in case of strong imbalances. |
| Fix the Institutions: promotes inclusive equality in careers through structural change in research organisations. | Structural change to institutional aspects of ULYSSEUS to promote the achievement of inclusive equality: how the project "speaks" to the outside, how it establishes new connections and new collaborations with other organisations respecting inclusive approaches. | When reading, viewing, or listening about the project and its activities, learn to observe the level of inclusiveness of language, visual images. When getting information about the ULYSSEUS partners, learn to search for information about the level of inclusiveness as described on the website and public documents. Be prepared, in case of moving to another university, to visit offices dedicated to EDI and ask for support or initiatives organised. |
| Fix the Knowledge (or "gendered innovations"): stimulates excellence in science and technology by integrating sex, gender, and intersectional analysis into research. | To introduce attention to EDI in performing research and organising teaching activities. Micro-actions may monitor that the new research projects and new teaching paths that will arise from ULYSSEUS include appropriate references to the themes of equality, diversity, and inclusion. Another possibility is to guide ULYSSEUS partners in identifying new partners already active in implementing inclusivity or interested in including activities to promote EDI in new proposals. | When attending training or courses observe to what extent references to gender, diversity and inclusion are made. Be prepared to ask for explanations in case no or insufficient references are made. |

Source: Own elaboration. Citations are included within the text and are linked to a reference containing the all the information needed to locate the original document.

**Table 3.** EDI micro-actions of the ULYSSEUS P-GAP.

| Objective | Macro Area | Goal Planned | Measure of Success | Benefit for Students/ESR/Staff/Community |
|---|---|---|---|---|
| Add gender/diversity/inclusion sensitive indicators and KPIs | Fix the Numbers | Select a harmonised set of KPIs and the gender diversity inclusion indicators to empower more organisations to ensure the equal treatment of women and men recording and monitoring the costs and outcomes of investing in policy change in the university | Data collected. Suggestions about ameliorating/keeping the situation balanced distributed to partners. | - To ensure representation of protected groups of staff is proportionate throughout all academic services<br>- To develop a more inclusive culture via more capable, inclusive leadership and management<br>- To ensure a proportionate representation of protected groups of staff and students |
| Add EDI-related criteria to impact and outcomes measurements | Fix the numbers Fix the knowledge | Training for applicants and reviewers, and the research community, to achieve greater equity, diversity, and inclusion in their research | Training organised and recorded | - To increase creativity, productivity, engagement, and innovation<br>- To stimulate diverse research personnel to provide a diverse set of role models who can mentor and activate students in different ways |
| Add qualitative and not only quantitative targets, with attention to gender and diversity | Fix the numbers | Seminars about qualitative and not only quantitative targets, with attention to gender and diversity Integrating these considerations into the policies, processes, excellence indicators and evaluation criteria | Document distributed via social media, newsletter, website | - To Improve foundations, ensure legal compliance, and tackle risks<br>- To Improve accountability, leadership and decision making<br>To Improve workforce equality data collection |
| Add, where possible "in a gendered perspective" or" in an EDI perspective", with examples, in training activities | Fix the knowledge | Training in integrating EDI considerations into the policies, processes, excellence indicators and evaluation criteria | Training organised and recorded | - To attract and retain a diverse student population<br>- To make sure all students can thrive and reach their full potential<br>- To eradicate prejudice and discrimination based on an individual or group of individuals' protected characteristics<br>- To implement a comprehensive training plan for diverse trainees for increasing the pool of diverse talent, at the same time enhancing pathways for growth and the likelihood of retention |
| Educate about gender equality in the ULYSSEUS and wider community | Fix the knowledge | Prepare a MOOC about Gender Equality | MOOC created and launched | - To build awareness and education around EDI |
| Show how attention to gender and EDI will be required (and monitored/evaluated) in future projects/activities, plan guidelines on how to reach this aim (guidelines to be widely distributed) | Fix the institution (the project) | Seminars about integrating these considerations into the policies, processes, excellence indicators and evaluation criteria | Training organised and recorded. Document distributed via social media, newsletter, website | - To improve accountability, leadership and decision making |
| Address gender-based violence in academia | Fix the organisation Fix the knowledge | Organise a presentation of the UniSAFE EU funded project. "Making universities and research organisations safe from gender-based violence" https://unisafe-gbv.eu (accessed on 1 October 2022) | Presentation organised | - To improve mechanisms for addressing unacceptable behaviour |

Source: Own elaboration. Citations are included within the text and are linked to a reference containing the all the information needed to locate the original document.

(2) The students will be involved also in activities conducted in the P-GAP created in the Gender Equality Agenda by ULYSSEUS. The activities aimed at EDI themes had to meet the European Commission's requirements for all projects funded under the European Union framework programs for research and innovation. In particular, the research study followed the three major strategic approaches to gender equality in science research, policy,

and practice adopted by governments and universities over the past several decades, as described and recommended by Prof. Londa Schiebinger (https://genderedinnovations.stanford.edu/what-is-gendered-innovations.html, accessed on 1 October 2022), adapting them for the European University Alliance:

All micro-actions, and their measures of success, will be specific, measurable, achievable, realistic, and time-bound (SMART) [50] following the European Institute for Gender Equality (EIGE) recommendations to facilitate the successful implementation of objectives, targets, and measures for Gender Equality Plans. The four mandatory process-related requirements (building blocks) and the five recommended thematic areas were considered, when possible, in choosing the micro-actions.

What follows is an extract of the table summarising the EDI micro-actions of the ULYSSEUS P-GAP. A column has been added to the right to specify the benefits for the students and ESRs.

## 8. Discussion and Conclusions

Over the past two decades, equality, diversity, and inclusion (EDI) has become increasingly important in universities and in the workplace at large. This sensitivity to EDI issues has led to an increase in the number of initiatives, working groups, policies, reports, and EU initiatives [51]. Despite these efforts, progress, especially in educational institutions, is still slow and strenuous.

While this study was successful, there were two limitations. A first limitation is that the various micro-projects proposed are embedded in a strategy similar to a GEP, concerned with continuing to advance gender equality and not focused on addressing EDI regarding post-university work opportunities for students.

While initiatives directly aimed at students can potentially have a more significant impact, currently, the six universities are not planning to organise more focused interventions. Therefore, the choice to widen micro-actions to address EDI (and not merely gender, including where appropriate intersectionality) is nonetheless a stimulus for students and favours them acquiring familiarity with behaviours and approaches that the students will look for and apply in their future working life.

Another limitation is that the implemented initiatives do not refer formally and officially to higher education-to-work transition. The micro-actions involve students and show them the benefits of and EDI approach and its usability in any work context. However, the micro-actions are related to ULYSSEUS and its implementation. During the micro-actions, references to the expendability of the acquired knowledge and soft skills in a working environment are made and remain explicit in the outputs (documents, slide presentations, publications). However, the students do not join the activities to facilitate study-work transition.

Nevertheless, this would open the way to the usefulness of formalising the most successful micro-actions in the future, implementing a formal pathway inspired by the micro-actions, according to the available resources. However, an indirect advantage of the lack of direct reference to the work-life transition could be that of attracting students who would not have followed a formal path.

Bearing in mind the limitations mentioned above, the approach chosen in this research study opens new possibilities for action and research, going beyond the findings of previous studies. Currently, there is no direct link between the importance of valuing and incrementing inclusion, raising awareness of equality, diversity, and inclusion (EDI) strategies and policies, and the transition from HE to work, as shown in the literature review. This paper contributes to filling this gap through a case study in which micro-actions related to EDI are enacted at the international level in a European University Alliance (EUA) through an innovative strategy unseen in other EUAs. The choice of a new strategy not directly related to facilitating young people's transitions from education to employment is due to the awareness of the importance of using any available strategy to address the topic. Other academic initiatives address the HE-to-work transition directly but adding

new perspectives and strategies may help understand EDI's importance in working and life contexts throughout life, further enhancing students' employability competencies and integrating current strategies aimed at other skills [25].

The post-COVID-19 pandemic academic environment is still uncertain due to numerous economic, social, and medical factors. As shown in previous sections of the article, the COVID-19 impact on youth employment is well documented, as well as the impact on the most fragile segments of the population. In this context, an awareness of the importance of addressing EDI-related topics, and a set of practical experiences in recognising critical or invisible potential grounds for discrimination, as offered by participating in the P-GAP, can have positive outcomes. In particular, this research shows the potential to better manage the students' study-to-work passage and acquire the capability of reading potential difficulties in the broader set of phenomena, thus becoming more resilient and adding a new strategy that could be complementary to the advisory group studied by Sobrany et al. [36].

The conceptualisation of students as "being in transition" [39] will be even more evident to the students themselves and the administrative and teaching academic staff once the European degrees that should become one of the outcomes of the European Universities Initiative become a reality. Thus, another dimension will be added to the feeling of being in transition, as described by Brooks et al. Being in transition also geographically and culturally among the partner universities to obtain a degree will add further dimensions to the following transition from HE to work. The P-GAP, enacted in the six universities of ULYSSEUS, could create a common ground, leading to better EDIs policies and strategies, thanks to the inevitable constructions and localisations adopted in the six countries involved in ULYSSEUS.

Alger defines the journey from access and diversity to genuine inclusion as a marathon rather than a sprint, where "small victories along the way are acknowledged and celebrated so as to provide continuing momentum and optimism for the long road ahead" [18] (p. 72). The micro-actions implemented in ULYSSEUS, in years when physical mobility is extremely limited, and therefore the interactions and collaborative work happen mainly online, reinforce Alger's vision and may represent small victories for the students involved and favour their future desire and substantial contribution in transferring the inclusive experience made in their academic years to their working environments or looking for inclusivity practices to apply in the working organisations.

To complement the work of this study, it would be interesting to explore further the following questions and statements.

It is often overlooked that there are fundamental flaws in the way EDI has developed in higher education. Some questions are still relevant: why does EDI, in its current form, allow systems of injustice to persist? How can universities break out of the cycle of performativity?

First and foremost, EDI is in fact not the answer, but rather the beginning of a process that needs continuous and rigorous monitoring. Work labelled as EDI focuses, very often, solely on supporting diversity efforts. In academies for example, it is limited to increasing the representation of vulnerable and/or underrepresented groups in various roles but does not stimulate exploration of the ways in which current work structures and practices allow injustice to flourish.

One example is the language used. Using words with different meanings as synonyms, such as "equality" and "equity", "diversity" and "inclusiveness". This confusion of terminology creates a false sense of individual and institutional progression and helps, intentionally or not, to mask systemic injustices.

The acronym "EDI" itself is a nebulous term that brings together three distinct ideas, "diversity", "equality", and "inclusivity", linked in complex and articulate ways that require deep reflection in their application in the contemporary educational and social context. Treating EDI as a complete package and not as separate entities risks leading to the erroneous belief that respecting diversity will automatically guarantee "inclusion" and "equality" and vice versa.

Is academia really exploring the ways in which inequalities are actively reproduced in our institutions?

Is academia moving not to make a pure tick box exercise, an aesthetic solution to a deeply rooted problem?

Increasing the representation of minority groups demonstrates an effective commitment to change, but this is not enough; academia needs to be open to learning more from community relationships, both inside and outside the institution, and to initiate an effort that is not individual, but community based.

The proposed reflective journey within the academy starts with looking beyond the idea that EDI is enough, to come to see it not as the goal or the destination, but as a tool for recognising the power that fuels oppressive structures. The outcome could thus contribute to a life-wide perspective, applicable in any working environment, for a more inclusive Europe in the long term.

**Author Contributions:** Conceptualization, A.S., C.L. and R.B.; writing—original draft preparation, R.B. and A.S.; writing—review and editing, A.S., C.L., and R.B. All authors have read and agreed to the published version of the manuscript.

**Funding:** This research was funded by COMPASS project, under the Agreement No. 101035809.

**Institutional Review Board Statement:** Not applicable.

**Informed Consent Statement:** Not applicable.

**Data Availability Statement:** Not applicable.

**Acknowledgments:** This work has been partly supported by ULYSSEUS EU Alliance Project under the Agreement No. 101004050.

**Conflicts of Interest:** The authors declare no conflict of interest.

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
