# Peer review of "Equality, Diversity, and Inclusion Strategies Adopted in a European University Alliance to Facilitate the Higher Education-to-Work Transition"

_societies, doi:10.3390/soc12050140_

Round 1

Reviewer 1 Report

This paper studies equality, diversity, and inclusion (EDI) strategies and policies related to one of the new European University Alliances called “ULYSSEUS” that involves six universities from four European regions. The paper is very topical and policy relevant as it looks at the important theme of EDI as well as the case of the new European University Initiative. While the paper is insightful regarding these topics, in the current version it sometimes reads more like an institutional report than a scientific paper. The theory, methods, and data sections could be elaborated more to guide the empirical analysis. The link between EDI strategies and transitions from higher education to work, which is flagged in the paper, would benefit from more elaboration at the conceptual level. The respective link to the mentioned impact of the Covid-19 pandemic could also be further specified. It is helpful that the literature review identifies gaps in the literature, but it would be good to then revisit these more explicitly in terms of how the analysis contributes to filling these gaps. The paper lacks a substantial discussion of the empirical findings. The selection of the “ULYSSEUS” alliance case and, within this alliance, the focus on the University of Genoa could be better explained. Also, the relationship the authors have to the case analysed is not entirely clear. On page 6 it is written that “In ULYSSEUS, the authors try to promote EDI inspired by strategies connected with the 267 European values enshrined in Article 2 of the Treaty on European Union: pluralism, tolerance, justice, solidarity, non-discrimination, and equality.” This could imply that the authors are involved in the implementation of the case that they analyse, which, in turn, might have methodological implications. In sum, I find that this is a promising paper on a very relevant topic, but that at this point its different elements do not yet fit together sufficiently well.

Author Response

While the paper is insightful regarding these topics, in the current version it sometimes reads more like an institutional report than a scientific paper.

We hope the numerous integrations added have solved this problem.

The theory, methods, and data sections could be elaborated more to guide the empirical analysis.

The sections have been enriched.

The link between EDI strategies and transitions from higher education to work, which is flagged in the paper, would benefit from more elaboration at the conceptual level.

The conceptual level has been enriched.

The respective link to the mentioned impact of the Covid-19 pandemic could also be further specified.

Links and comments have been added.

It is helpful that the literature review identifies gaps in the literature, but it would be good to then revisit these more explicitly in terms of how the analysis contributes to filling these gaps.

Stronger links between the literature and the discussion can be found in the last paragraph, renamed and entirely revised.

The paper lacks a substantial discussion of the empirical findings.

It is too early to have findings. The article is meant to describe the implementation of a process, not originally aimed at promoting a better work transition, but seen by the authors as a potential opportunity to indirectly show and stimulate practical experience on EDI, as a set of values and practices relevant in any career path.

The selection of the “ULYSSEUS” alliance case and, within this alliance, the focus on the University of Genoa could be better explained.

The authors work at the University of Genoa and are responsible for the implementation of WP5, where the initiative described is embedded. This information is not clearly stated in the paper.

Also, the relationship the authors have to the case analysed is not entirely clear.

We give more details on our role in the case study.

On page 6 it is written that “In ULYSSEUS, the authors try to promote EDI inspired by strategies connected with the European values enshrined in Article 2 of the Treaty on European Union: pluralism, tolerance, justice, solidarity, non-discrimination, and equality.” This could imply that the authors are involved in the implementation of the case that they analyse, which, in turn, might have methodological implications.

We give more details on our role in the case study.

In sum, I find that this is a promising paper on a very relevant topic, but that at this point its different elements do not yet fit together sufficiently well. 

We addressed all the criticisms raised and we hope that the article now has a stronger internal coherence.

Reviewer 2 Report

This submission has an interesting starting point in thinking about opportunities for a European Universities Initiative network to address EDI challenges arising from the pandemic. The authors supply data showing worrying trends in youth unemployment that such projects could address in working with students.

However, the paper does not follow up in addressing the topic, since much of the following discussion is about European Universities Initiative structures and requirements for gender equality action plans which only very loosely relate to the question. Indeed the various micro-projects proposed appear more concerned with continuing to advance gender equality than addressing EDI with respect to post-university work opportunities for students. 

The submission has a good idea but, in my judgement needs a good deal more work to develop an integrated argument with evidence to support the question. Perhaps as the micro-initiatives develop the authors can revisit their question and secure some data addressing the question effectively.

Author Response

As an aspect to be improved, I would recommend increasing the number of bibliographical references and, perhaps, improving the discussion of empirical studies on the subject.

The number of bibliographical references has been increased.

The discussion of empirical studies on the subject has been widened (this aspect has been requested also by Reviewer 1).

Reviewer 3 Report

The article raises a subject of great social, scientific and educational interest, fulfilling with rigour and consistency what is expected of a scientific article. My assessment is positive and its publication is recommended.

As an aspect to be improved, I would recommend increasing the number of bibliographical references and, perhaps, improving the discussion of empirical studies on the subject.

Author Response

… much of the following discussion is about European Universities Initiative structures and requirements for gender equality action plans which only very loosely relate to the question.

A comment has been added: Lines 69-74. The conclusions now include indirect references to this critique.

Indeed, the various micro-projects proposed appear more concerned with continuing to advance gender equality than addressing EDI with respect to post-university work opportunities for students.

See conclusions. Comments have been added to show how the gender related aspects are de facto widened to include diversity and inclusion.

Perhaps as the micro-initiatives develop the authors can revisit their question and secure some data addressing the question effectively.

The article describes the first steps of a process that will take at least two years to be implemented. It is too early to provide data. However, we deemed it important to stimulate a reflection on the potential “use” of European Universities also in promoting the students’ transition through activities not aimed initially at that scope.